

# Estimation of soil erosion considering soil loss tolerance in karst area

**Yue Cao[1,2,3], Shijie Wang[1,3], Xiaoyong Bai[1,3] , Huiwen Li[1,2,3], Cheng Zeng[1,3] , Yichao Tian[4] , Fei Chen[1,3] , Luhua Wu[1,2,3] , Mingming Wang[1,2,3]**

[1] State Key Laboratory of Environmental Geochemistry, Institute of Geochemistry, Chinese Academy of Sciences, 99
Lincheng West Road, Guiyang 550081, Guizhou Province, China.

[2] University of Chinese Academy of Sciences, Beijing 100049, China.

[3] Puding Karst Ecosystem Observation and Research Station, Chinese Academy of Sciences, Puding 562100, China.

[4] School of Resources and Environment of Qinzhou University, Qinzhou 535099, China.

*Correspondence to:* Xiaoyong Bai (baixiaoyong@126.com)

**Abstract.** The prediction of soil erosion is critical to regional ecological assessment and sustainable development. However, due to the geological background of the karst area, the soil holding capacity is very limited, so it is necessary to consider the allowable loss of soil. Here we took thermodynamic dissolution model of carbonate rocks and the lithological characteristics to estimate soil loss tolerance, and corrected and quantitatively evaluated the soil erosion. Major findings are as follows: 1) The soil loss tolerance of homogenous carbonate rocks is 31.10 $t \cdot ha \cdot yr^{-1}$, carbonate rock intercalated with clastic rocks is 120.81 $t \cdot ha \cdot yr^{-1}$, carbonate/clastic rock alternations is 282.55 $t \cdot ha \cdot yr^{-1}$, and clastic rock is 500 $t \cdot ha \cdot yr^{-1}$. 2) After the correction of the soil loss tolerance, the average annual amount of soil loss in the study area is 3.08 $t \cdot ha \cdot yr^{-1}$, which is 41.12% of the model. The predicted value of soil erosion is nearly the same as the observed value after modification. 3) It is necessary to reconsider the risk assessment model of soil erosion applicable to karst areas. This paper proposes an idea to estimate soil erosion based on the allowable loss of soil, which is more scientifically and accurately to reflect the soil erosion status of the study area compared with the traditional way. This study provides a corresponding reference for the formulation of soil and water conservation policies in China and the world's karst regions.

**Keywords:** karst; soil erosion; loss tolerance; carbonate; rock weathering

**1 Introduction**

Karst landscapes occupy approximately 12% of the continental terrain and have highly fragile environments (Febles-Gonzalez et al., 2012). Soil erosion and progressive degradation have been identified as severe geo-environmental hazards in many karst areas (Fernández and Vega, 2016; Hu et al., 2018; Parise et al., 2009). Against the background of rocky desertification and serious soil erosion in karst areas of southern China, the most important problem of ecological environment construction is the prevention and control of soil erosion. The effective control of soil erosion requires a profound understanding and grasp of soil erosion mechanism in karst areas (Ni et al., 2010). Therefore, research on soil erosion in karst areas should be considered in combination with the local geological environment(Zhang et al., 2013), and simulate the soil erosion in the area as far as possible.

At present, many scholars have carried out research on the special geomorphological features of karst areas. Vigiak assessed sediment fluxes in the Danube Basin from 1995 to 2009 with a SWAT (Soil and Water Assessment Tool) model and suggested that the model underestimations were correlated with the Alpine and karst areas in the basin(Vigiak et al., 2017). Smirnova estimated soil erosion within a karst sinkhole in the dry steppe subzone and conducted a quantitative assessment of pedodiversity(Smirnova and Gennadiev, 2017). Li and Zeng analysed the temporal and spatial evolution characteristics of soil erosion in an area with typical karst geomorphology in China and put forward a new description of soil erosion due to underground leakage (Li et al., 2015; Zeng et al., 2017). Aksoy simulated rainfall and erosion in a

karst rocky desertification area with an indoor laboratory experimental setup, and established an empirical model (Aksoy et al., 2017). López-Vicente presented a method to obtain accurate DEMs in the karstic endorheic catchments in the Spanish Pyrenees and provided a basis for calculating the topography factor in the RUSLE for an accurate assessment of the topographical and geomorphological features in karstic environments (Lopez-Vicente et al., 2009). These studies

provide a referential framework for the study on soil erosion in karst areas.

However, the karst rocks in the south China are dominated by carbonate rocks that are ancient, hard, pure and lacking soil cover, causing the karst soil formation rate to be very slow (Yuan, 1988). Mountainous karst soil is usually only ten to several dozen centimetres thick (Cao et al., 2008), and bedrock can even be exposed. Consequently, the earlier approaches did not consider the details of the hydrological and erosive processes that are controlled by the

conditions of karst development and may have overestimated the erosion rates.

Therefore, taking the karst area in South China as the research object, this paper combines with the lithological characteristics of the karst area to estimate soil loss tolerance and correct the prediction model of soil erosion, which can more closely reflect to the real conditions. On this basis, we could accurately estimate the soil erosion in ecologically fragile karst areas. Estimations that are more accurate would provide important theoretical information for determining

prevention and control measures for soil erosion, managing rocky desertification and regional sustainable development.





## 2 Materials and methods

### 2.1 Study region

The karst valley studied in this paper is in the karst area of southern China. The geographical location of the study area is 105°31' N to 13°47' N and 26° 23' E to 33°37' E, and the study area covers $2.86\times10^5$ km$^2$. This area accounts for 14.77% of southern China and is an important part of the karst region in southern China. The climate is subtropical monsoon, and the average annual rainfall is more than 800 mm. The soil types are mainly yellow soil and lime soil. The exposed karst in the study area accounts for 46.06% of the total area. Here, the karstification is strong, the soil is discontinuous, and the underground fissures and caves develop. Moreover, the widespread soil erosion leads to thinning of the soil layer and a fragile ecosystem.

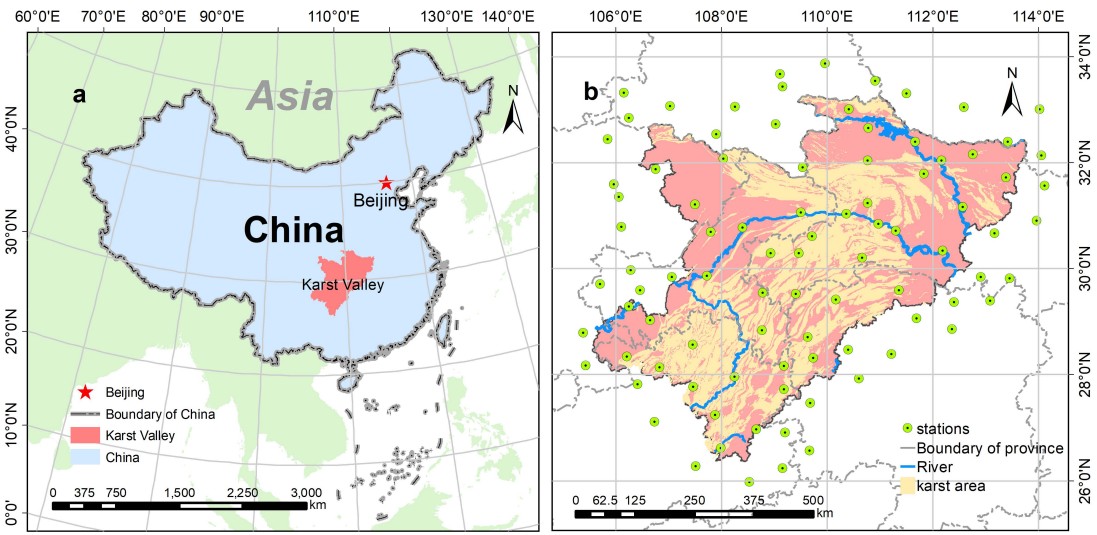

Fig. 1. Location of the study area in China (a) and the valley (b)




## 2.2 Data sources

Data sources are shown in Table 1. Daily meteorological data, including precipitation (P, mm) and

evapotranspiration (E, mm), in 90 stations in and around the valley in 2015 (station locations are shown on Fig.1-b).


Table 1. Data sources

| Data type | Data source | Website |
|-----------|-------------|---------|
| Vector boundary of valley area | Karst Science Data Center | http://www.karstdata.cn/ |
| Karst region of valley area | Karst Science Data Center | http://www.karstdata.cn/ |
| Lithologic map of valley area | Karst Science Data Center | http://www.karstdata.cn/ |
| DEM | Geospatial data cloud | http://www.gscloud.cn/ |
| NDVI | Geospatial data cloud | http://www.gscloud.cn/ |
| Daily meteorological data | China Meteorological Data Network | http://data.cma.cn |
| Land use | Resource and environment science data center | http://www.resdc.cn |
| Soil data | Resource and environment science data center | http://www.resdc.cn |

## 2.3 RUSLE model

The RUSLE (Revised Universal Soil Loss Equation) model (Renard et al., 1997) is a widely used soil erosion

prediction model, both in China and worldwide. The model has the advantages of concise structure, simple calculation,

strong practicability and comprehensive ability as well as parameters with definite physical meanings. In this study, the

RUSLE was used to evaluate the soil erosion in the study area. The basic form of the model is

$$A = R \times K \times L \times S \times C \times P \tag{1}$$

In the formula, $A$ is the amount of soil erosion in a year (t·hm$^{-2}$·a$^{-1}$), $R$ is the rainfall factor ((MJ·mm)/(hm$^2$·h)), $K$





is the soil erodibility factor $((t \cdot km^2 \cdot h)/(km^2 MJ \cdot mm))$, $LS$ is the topographic factor, $C$ is the vegetation cover factor and

$P$ is the conservation practice factor. The index values of each factor in the RUSLE model are obtained by remote sensing

and GIS technology.

Rainfall erosivity (R) is an important index that is used to evaluate soil erosion and transport erosion from the

rainfall, reflecting the potential for rainfall to cause soil erosion. Zhang compared the five methods of daily, monthly

and annual rainfall, and concluded that the daily rainfall model has the highest accuracy in calculating rainfall erosivity

(Zhang and FU, 2003). The calculation methods are as follows:

$$R_i = \alpha \sum_{j=1}^{K} (D_j)^{\beta} \tag{2}$$

In this formula, $R_i$ is the value of a half months $(MJ \cdot mm \cdot / (hm^2 \cdot h))$, $K$ is the number of days in the half month period. $D_j$

is the erosive daily rainfall of j days in half a month. $D_j$ is the erosive daily rainfall of the $j$ th day in half month period,

which requires 12 mm daily rainfall. Because of the development of underground fissures and pipelines, there is leakage

in karst area, so the erosive rainfall standard is 30 (mm) (Wei, 2011), otherwise it is calculated by 0. $\alpha$ and $\beta$ are

undetermined parameters of the model.

$$\beta = 0.8363 + \frac{18.144}{P(d)} + \frac{24.455}{P(y)} \tag{3}$$

$$\alpha = 21.586 \beta^{-7.1891} \tag{4}$$

In this formula, $P(d)$ is the daily mean rainfall of daily rainfall greater than 12 (no-karst) or 30 (karst area) (mm).

$P(y)$ is the annual mean rainfall of daily rainfall greater than 12 (no-karst) or 30 (karst area) (mm).



Soil erodibility (K) is an important index to evaluate the sensitivity of soil to erosion. In this study, K were estimated

by Williams in the EPIC model in 1990(Williams, 1990), and the calculation formula is

$$K = \left\{0.2 + 0.3\,exp\left[-0.0256SAN\left(1 - \frac{SIL}{100}\right)\right]\right\} \times \left(\frac{SIL}{SIA + SIL}\right)^{0.3}$$

$$\times \left[1 - \frac{0.25C}{C + exp(3.72 - 2.95C)}\right] \times \left[1 - \frac{0.7SN}{SN + exp(22.9SN - 5.51)}\right] \qquad (5)$$

In this formula, $K$ is the value of soil erodibility, and the unit is (t·acre·h)/(100·acre·ft·tanf·in); for international units,

this unit needs to be multiplied by the conversion coefficient 0.1317 to transform it to (t·km$^2$·h)/(km$^2$MJ·mm). *SAN*, *SIL*,

*CLA* and *C* are the sand (0.050~2.000 mm), silt (0.002~0.050 mm), clay (<0.002 mm) and organic matter contents, where

*SN*=1-*SAN*/100.

The slope length factor and slope factor reflect the effect of the topographic and geomorphic characteristics on the

soil erosion. The S factor (slope factor) is the ratio of the soil erosion per unit area for an arbitrary slope and the amount

of soil erosion per unit area under the same slope. According to the research of Mccool and Liu(Mccool et al., 1987)

(Liu et al., 2000), the S factor is usually calculated in stages, and the formula is

$$S = \begin{cases} 10.8\,sin\,\theta + 0.03 & \theta < 5° \\ 16.8\,sin\,\theta - 0.5 & 5° \le \theta < 10° \\ 21.9\,sin\,\theta - 0.96 & \theta \ge 10° \end{cases} \qquad (6)$$

where $S$ is the slope factor and $\theta$ is the slope (°).

The common formula for the slope length factor is as follows(Mccool et al., 1989):

$$L = (\lambda/22.13)^m \qquad (7)$$





$$m = 0.2 \quad \theta \leq 1°$$

$$m = 0.3 \quad 1° < \theta \leq 3°$$

$$m = 0.4 \quad 3° < \theta \leq 5°$$

$$m = 0.5 \quad \theta > 5°$$

In the slope length factor formula, $L$ is the slope length factor and $\lambda$ is the slope length value (m), which is the

projection distance of the runoff source to the depressions or grooves along the line direction. $m$ is the slope length index,

which is varied to describe different slopes.

The vegetation coverage factor characterizes the combined effect of all the vegetation characteristics on the soil

erosion. We use the NDVI from MODIS and the method of Cai (Cai et al., 2000); the formula is


$$C = \begin{cases} 1 \\ 0.6508 - 0.3436 lgf \\ 0 \end{cases}$$

$$f = \frac{NDVI - NDVI_{min}}{NDVI_{max} - NDVI_{min}} \tag{8}$$

In the vegetation coverage factors formula, $C$ is the vegetation coverage factor, $f$ is the ratio of vegetation coverage,

and $NDVI_{max}$ and $NDVI_{min}$ are the maximum and minimum of the NDVI in study area.

The soil and water conservation measure factor refers to the ratio of the soil loss under specific soil and water

conservation measures to the amount of soil loss on the slope where the measures were not implemented. For this factor,

we refer to Xu 's assignment of the different types of land use(Table 2) (Xu et al., 2011).

Table 2. Soil and water conservation factors in valley



| Land use types | Forest | Grassland | Cropland | Paddy field | Town | Road | Waters | Unused land |
|---|---|---|---|---|---|---|---|---|
| p | 1 | 1 | 0.4 | 0.15 | 0 | 0 | 0 | 1 |

## 2.4 Determination of the soil loss tolerance

The worldwide soil loss is determined from the rate of soil formation, the normal crop requirements and the

characteristics of the soil. This method is widely used in the Loess Plateau of China and the black soil area in northeastern

China but is not suitable for the karst area in southern China. Because the soils in the karst area are mainly derived from

the local bedrock, and the amount of allowable soil loss largely depends on the rate of soil formation under specific

environmental and geological conditions (Sun et al., 2014; Wang et al., 2004; Zhou et al., 2009). When the rate of soil

erosion is greater than the rate of parent rock exposure, the soil will lose its parent material, possibly after a few years,

causing soil graveling and rocky desertification. In contrast, if the rate of soil loss is less than the rate of parent rock

exposure, the total amount of soil theoretically increases annually. Therefore, according to the "short board theory" of

cask water holding, the rate of soil loss should be relatively balanced with the rate of soil formation to maintain a stable

soil fertility and land productivity. At present, many scholars (Cao et al., 2008; Li et al., 2017; LI et al., 2006) in China

have directly determined the soil loss tolerance in a carbonate area according to the rate of soil formation; this method

is also common worldwide.

The karst pedosphere system is divided into inputs and outputs. The input system includes rock weathering,

atmospheric dust, and biological return, all of which are transported to the inner layer of soil. The output system includes

chemical loss, physical loss and biological loss, all of which are lost from the soil layer. When the system reaches a



stable equilibrium, there is a balance:

$$W_i + F_i + B_i = C_0 + P_0 + B_0 \tag{9}$$

Where $W_i$ is the soil weathering rate of the rock (t·km$^{-2}$·a$^{-1}$), $F_i$ is the input rate of atmospheric precipitation and dust fall (t·km$^{-2}$·a$^{-1}$), $B_i$ is the biological return input rate (t·km$^{-2}$·a$^{-1}$), $C_o$ is the output rate of the chemical loss (t·km$^{-2}$·a$^{-1}$), $P_o$ is the physical loss output rate (t·km$^{-2}$·a$^{-1}$), and $B_o$ is the outflow rate of the biological loss (t·km$^{-2}$·a$^{-1}$).

According to Bai's study(Bai and Wang, 2011), the rates of chemical loss and atmospheric dust fall in the karst region of South China may vary from 2.56 to 4.44 t·km$^{-2}$·a$^{-1}$ but are basically equal. The biogenic soil is negligible relative to the weathering of carbonates by acid-insoluble material; therefore, it is theoretically conceivable that the rates of chemical loss and atmospheric dust fall are approximately balanced. Due to the dissolution of rocks in the karst region, there are many pores, fissures and pipe holes in the study area. The physical loss of soil includes surface loss and underground leakage (Zhang et al., 2009). The rate of surface loss in the karst area is the allowable loss of surface soil equal to the rock weathering rate minus the rate of underground runoff:

$$P_s = W_i - P_u \tag{10}$$

where $P_s$ is the rate of surface runoff and $P_u$ is the rate of underground runoff.

The rate of surface loss in the karst area is equal to the rate at which the rock is weathered into soil, minus the rate of the underground runoff, and is the allowable loss of surface soil in the karst area. Many long-term field monitoring records show that the dissolution rate of limestone is 1.5 to 2 times faster than that of dolomite. The allowable loss of



clastic rocks (equivalent to the rate of formation) is calculated as 500 t·km$^{-2}$·a$^{-1}$. In the continuous carbonate area where underground rivers are widely developed, the rate of underground loss is 13.6 t·km$^{-2}$·a$^{-1}$(Zhang et al., 2009). Various types of carbonate rock have been studied by LI (LI et al., 2006). Based on this work, the formula of soil loss tolerance for different rock types in karst areas is as follows:

$$W_i = v \cdot \rho \cdot Q \cdot M + N \cdot (1 - M) \tag{11}$$

where $v$ is the carbonate dissolution rate (mm·a$^{-1}$ converted into t·km$^{-2}$·a$^{-1}$), $Q$ is the acid insoluble content (%), $M$ is the carbonate content (%), $\rho$ is the bulk density of the carbonate rock (t/m$^3$), and $N$ is the rate of the non-carbonate rock formation (t·km$^{-2}$·a$^{-1}$).

White provided a general equation for the dissolution of calcite at equilibrium (White, 2011):

$$CaCO_3 + CO_2 + H_2O \rightleftharpoons Ca^{2+} + 2HCO_3^- \tag{12}$$

Based on this equilibrium reaction, Gombert assumed that the carbonate area reached a dissolution equilibrium under local water, temperature and CO$_2$ conditions and created a thermodynamic dissolution model for carbonate areas as follows(Gombert, 2002):

$$D_{max} = 10^6(P - E)[Ca^{2+}]_{eq} = 10^6(P - E)(K_s K_1 K_0 / 4K_2\gamma_{(Ca^{2+})}^3)^{1/3}(pCO_2)^{1/3} \tag{13}$$

where $Dmax$ is the maximum dissolution rate of carbonate rocks under this equilibrium reaction, $P$ and $E$ are the total amounts of rainfall and evapotranspiration, $K_s$ is the calcite solubility product constant, $K_1$ is the equilibrium constant of CO$_2$ hydration and dissociation to HCO$_3^-$, $K_0$ is the equilibrium constant of CO$_2$ dissolved in water, $K_2$ is the equilibrium



constant of $CO_3{}^{2-}$, $\gamma_{Ca^{2+}}$ is the activity coefficient of $Ca^{2+}$ in solution, and $pCO_2$ is the partial pressure of $CO_2$ in the

soil or aquifer.

According to the work of Plummer (Plummer and Busenberg, 1982), $K_s$, $K_1$, $K_0$, and $K_2$ are functions of temperature

$T_k$ (K):

$$log(K_s) = -171.91 - 0.078T_k + 2839.32/T_k + 71.59log(T_k) \tag{14}$$

$$log(K_1) = -356.31 - 0.061T_k + \frac{21834.37}{T_k} + 126.8339lo\,g(T_k) - 1684915/T_k{}^2 \tag{15}$$

$$log(K_2) = -107.89 - 0.033T_k + \frac{5151.79}{T_k} + 38.93lo\,g(T_k) - 563713.9/T_k{}^2 \tag{16}$$

$$K_0 = 1.7 \times 10^{-4}/K_1 \tag{17}$$

The ionic activity coefficients of $Ca^{2+}$ and $HCO_3{}^-$ can be calculated by the Debye-Hückel equation (Plummer and

Busenberg, 1982):

$$log(\gamma_i) = -AZ_i{}^2 \frac{\sqrt{I}}{1+Ba_i\sqrt{I}} \tag{18}$$

$A$ and $B$ depend on the temperature T (°C), $a_i$ is the ionic radius(Dreybrodt, 1988), $Z_i$ is the ionic charge number, $I$

is the ionic strength, and $C_i$ is the ionic concentration (mol/L).

$$A = 0.4883 + 8.074 \times 10^{-4}T \tag{19}$$

$$B = 0.3241 + 1.6 \times 10^{-4}T \tag{20}$$

$$I = \frac{1}{2}\sum_i Z_i{}^2 C_i \tag{21}$$

The partial pressure of $CO_2$ in the soil or aquifer is calculated from the Brook formula(Brook et al., 2010):





$$log(pCO_2) = -3.47 + 2.09 \times (1 - e^{-0.00172E}) \tag{22}$$

In theory, each dissolved mol of $CaCO_3$ consumes one mol of $CO_2$. Therefore, the carbon sink fluxes (CSF) can be

calculated by the following equation (Li et al., 2018):

$$\because CSF = 10^6(P - E)[HCO_3^-]_{eq}/2 = 10^6(P - E)[Ca^{2+}]_{eq} \tag{23}$$

$$\therefore CSF = 10^6(P - E)(K_sK_1K_0/4K_2\gamma_{Ca^{2+}}\gamma_{(HCO_3^-)}^2)^{1/3}(pCO_2)^{1/3} \tag{24}$$

## 3 Results

### 3.1 Estimation of soil erosion based on RUSLE

Based on the RUSLE model, we estimate the amount of soil erosion and classify the study area. The results of

rainfall erosivity (R) factor, soil erodibility (K) factor, topographic (LS) factor, vegetation cover (C) factor and

conservation practice (P) factor show in Fig 2 (a-e). Then the amount of soil erosion in a year (A) is calculated (Fig2-f),

and the average value is 7.50 t·ha·yr$^{-1}$. There are nearly 70% of the area is micro-erosion, followed by mild, moderate,

strong, pole strong and violent erosion.



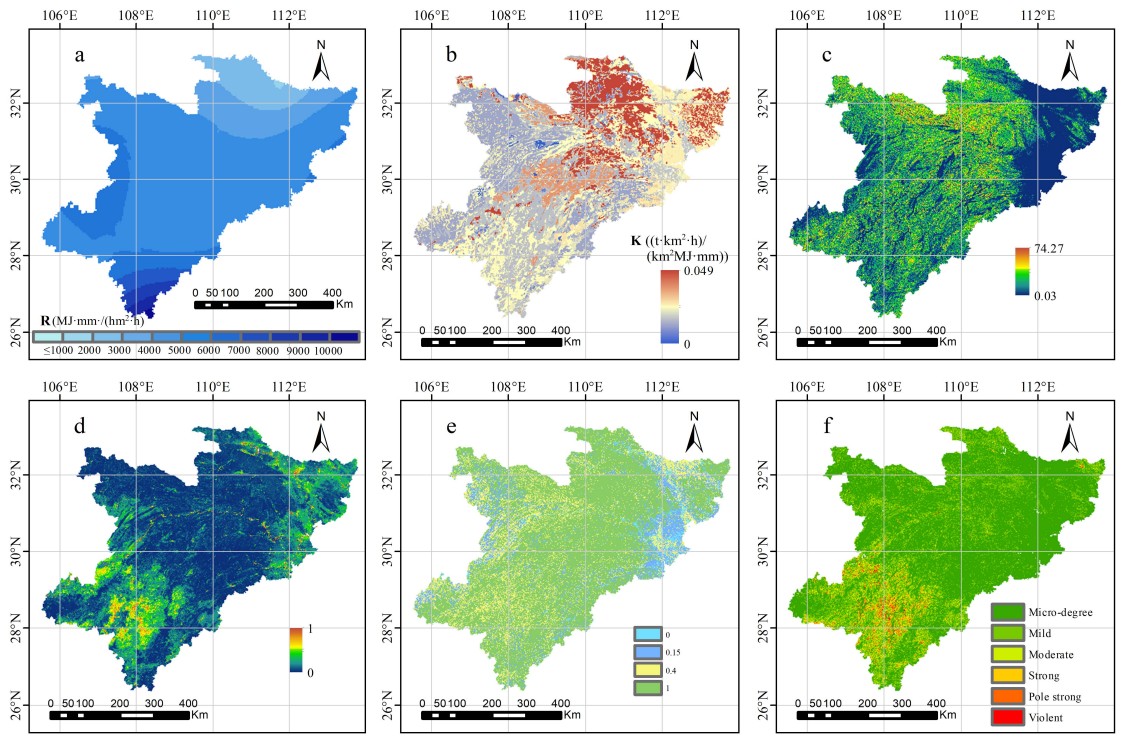

Fig. 2. The spatial distribution of factors in the RUSLE model

## 3.2 Correction with the soil loss tolerance

Compared with non-karst areas, karst areas with widespread carbonate rocks have low soil formation rates and less

soil. Therefore, the actual amount of soil erosion in this type of karst area will not exceed the allowable loss of soil.

Based on the spatial distribution of lithology in a karst region, according to the rate of carbonate rock dissolution and

non-carbonate rock formation, we obtain the soil loss tolerance in the karst region of the study area (Fig. 3-a). The results

show that the allowable loss of carbonate rock soils with different lithology combinations is different. The soil loss

tolerance of homogenous carbonate rocks is 27.73~33.47 t·ha·yr$^{-1}$, carbonate rock intercalated with clastic rocks is

109.44~128.64 t·ha·yr$^{-1}$, carbonate/clastic rock alternations is 264.42~294.43 t·ha·yr$^{-1}$, and clastic rock is 500 t·ha·yr$^{-1}$.

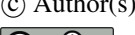



In addition, the amount of soil erosion in a year in the study of karst area is 1.34 t·ha·yr[-1]. However, the average annual

loss of soil erosion is 7.50 t·ha·yr[-1], which is considerably greater than the tolerance in the area. Therefore, it is necessary

to perform correction based on the result of model. The spatial distribution is shown in Fig. 3-b.

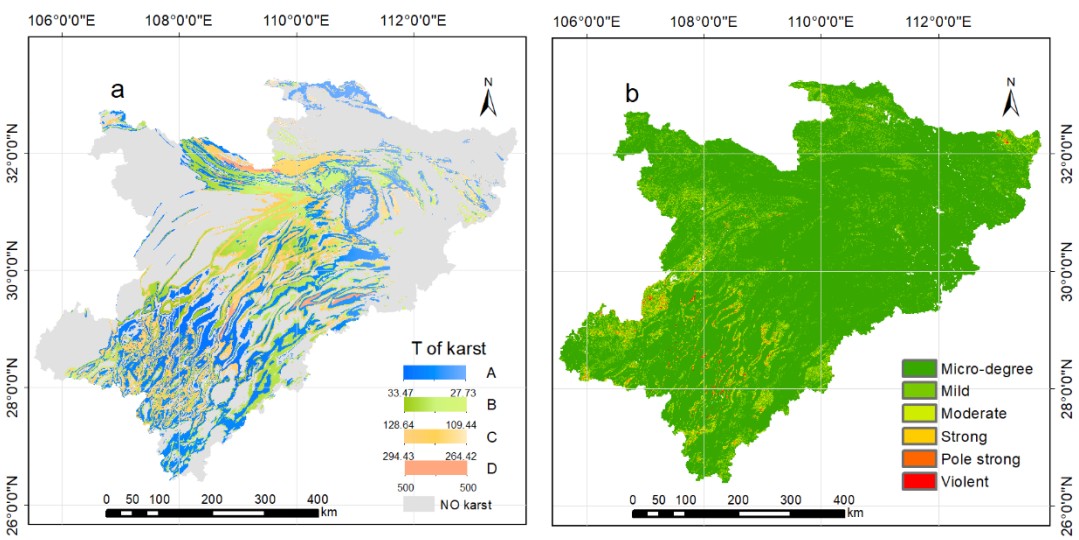

Fig. 3. The soil loss tolerance (a) and the correction of A (b)

((a) shows the amount of soil loss tolerance in area of homogenous carbonate rocks (A), carbonate rocks intercalated with clastic rocks (B), interbedded carbonate and clastic rocks (C) and clastic rocks (D))

### 3.3 Comparison with RUSLE

After the correction of the soil loss tolerance, the average annual amount of soil loss in the study area is 3.08 t·ha·yr[-]

[1], which is 41.12% of the RUSLE model. The average annual amount of soil loss in the study karst area is 1.34 t·ha·yr[-]

[1], which is 15.63% of the estimation of the model, which means there is an 84.37% overestimate. We compare the

correction with the RUSLE, and the results are followed (Table 3.).

Table 3. The soil erosion estimates for different erosion levels in the valley (A: RUSLE; B: the correction of the soil loss tolerance)



|  |  | Micro-degree | Mild | Moderate | Strong | Pole strong | Violent |
|---|---|---|---|---|---|---|---|
| Erosion area ($\times 10^3$t·yr$^{-1}$) | A | 197.29 | 67.40 | 13.35 | 4.89 | 2.79 | 0.62 |
|  | B | 240.86 | 34.83 | 6.06 | 2.06 | 1.17 | 0.32 |
| Area ratio (%) | A | 68.90 | 23.54 | 4.66 | 1.71 | 0.97 | 0.22 |
|  | B | 84.42 | 12.21 | 2.12 | 0.72 | 0.41 | 0.11 |
| Average modulus (t·ha·yr$^{-1}$) | A | 1.50 | 11.05 | 34.83 | 62.33 | 104.36 | 200.66 |
|  | B | 1.22 | 10.85 | 34.66 | 62.27 | 104.57 | 210.33 |
| Total soil loss ($\times 10^7$t·yr$^{-1}$) | A | 2.96 | 7.45 | 4.65 | 3.05 | 2.91 | 1.24 |
|  | B | 2.95 | 3.78 | 2.10 | 1.28 | 1.22 | 0.66 |
| Erosion ratio (%) | A | 13.29 | 33.47 | 20.90 | 13.70 | 13.08 | 5.56 |
|  | B | 24.56 | 31.49 | 17.51 | 10.70 | 10.21 | 5.53 |

It can be seen from Table 3 that due to the correction of the soil loss tolerance, the erosion area of the micro-degree

level increased, and the areas of mild, moderate, strong, pole strong and violent erosion reduced. In addition, the total

amount of erosion at each level decreased. The average modulus of micro, mild, moderate and strong erosion reduced

by a small amount. Although the total amount of erosion is reduced, the erosion area reduced by almost half, resulting

in a small increase in extremely pole strong and violent erosion. For the total amount of erosion, the corrected result is

46.62% of the RUSLE result, which is overestimated by 53.38%.

## 4 Discussion

### 4.1 Validity of the modifications

The carbonate rocks in the karst area of southern China are inherently deficient in soil-forming materials, and the

rate of soil formation is low. Soil erosion develops to a certain stage of "soilless flow". Therefore, the actual amount of

soil erosion in the karst area will not exceed the allowable loss of soil. It is necessary to revise the methodology again

based on the soil loss tolerance in the karst area to precisely calculate the of soil erosion. Therefore, this paper is based

on the spatial distribution of the lithology in the karst region; according to the carbonate dissolution rate and the rate of

non-carbonate rock formation, we obtain the soil loss tolerance in the karst area of the study area. The soil erosion

modulus is corrected, improving the accuracy of the result. This paper predicts the soil erosion modulus and establishes

a more suitable way for studying soil erosion in karst areas. In addition, this paper is of great significance to the

sustainable development of the society and economy of the karst areas in southern China and the world.

It is noteworthy that, according to the results of this paper, the allowable loss of soil in the karst valley area is 0.28

to 5 $t \cdot ha \cdot yr^{-1}$, which describes micro-degree erosion in the existing soil erosion classification standards. At the same time,

there is little soil in karst area. Once the soil erosion occurs, the risk is very high. This standard clearly does not conform

to the situation in the study area. Perhaps we can consider using the ratio of theoretical soil erosion to soil loss tolerance

to reflect the risk level of soil erosion in karst areas. That is to say, if the theoretical erosion is less than the tolerance,

even if the theoretical erosion is larger, it is still safe; if the theoretical erosion is greater than that, even if the theoretical

erosion is very small, it is also very dangerous.

**4.2 Comparison with the observed values**

We compare the results of this study with those of similar study areas (Table 4), . According to our study, the average

amount of soil erosion in a year in the study karst area is 1.34 $t \cdot ha \cdot yr^{-1}$. The average allowable loss of soil in the





homogenous carbonate area is 0.31 t·ha·yr$^{-1}$, in carbonate rocks intercalated with clastic rocks is 1.21 t·ha·yr$^{-1}$ and in

interbedded clastic rocks is 2.83 t·ha·yr$^{-1}$. Most of the measured small watersheds are typical karst areas, which are

concentrated in the homogenous carbonate area. The experimental soil erosion modulus is generally 0.4 t·ha·yr$^{-1}$, which

is consistent with our prediction. The study also shows that the soil erosion intensity is affected by the internal structure

of the lithology and that the greater the proportion of clastic rock is the greater the modulus of soil erosion.

Table 4. Comparison with the measured results of the adjacent study areas

| The first author | Study area | Time scale | Method | Soil erosion modulus |
|---|---|---|---|---|
| Chen (Chen, 1997) | Xichou peak cluster, Yunnan | 1997 | Experimental | 3.88 |
| Peng et al. (Jian and Ming-de, 2001) | Huajiang Karst gorge, Guizhou | 2000 | Piling | 0.16-8.44 |
| | | | Settling basin | 0.25 |
| Long et al. (Ming-zhong et al., 2014) | Huajiang Karst gorge, Guizhou | 2010 | Runoff plots | 0.05-0.29 |
| Wei (Wei, 2011) | Nanchuan karst valley area, Chongqing | 2009 & 2010 | Runoff plots | 0.07-0.41 |
| | | | $^{137}$Cs monitoring | 9.19-14.30 |
| This paper | Karst valley, South China | 2015 | Model improvements | 0.28-5.00 |

## 4.3 Prospects for future research

While new understandings and discoveries have been obtained, there are still some shortcomings. This paper studies

and calculates the amount of soil erosion in one year, during 2015; this timescale has some limitations. Future studies



will combine several periods of soil erosion data to study the temporal variations in the erosion in the karst valley and

then predict soil erosion status in the future. These studies will provide reference data and scientific support for the

development of soil and water conservation work and for the improvement of regional ecological and poverty problems

in karst regions.

**5 Conclusions**

Common soil erosion models usually do not consider the dual structure of karst areas, which often makes the

calculated soil erosion magnitude incorrect in karst areas. Moreover, the rate of soil formation in the karst areas is

relatively slow due to the restriction of the karst environment. Hence, the actual magnitude of soil loss tolerance should

be calculated by the soil formation rate of an area with carbonate rocks. The allowable loss of carbonate rock soils with

different lithology combinations is different. The soil loss tolerance of homogenous carbonate rocks is 31.10 $t \cdot ha \cdot yr^{-1}$,

carbonate rock intercalated with clastic rocks is 120.81 $t \cdot ha \cdot yr^{-1}$, carbonate/clastic rock alternations is 282.55 $t \cdot ha \cdot yr^{-1}$,

and clastic rock is 500 $t \cdot ha \cdot yr^{-1}$.

When calculating the soil erosion, it is necessary to determine the soil formation rate from the lithology of the karst

area; based on this information, the preliminary estimates of soil erosion that are calculated by the soil loss tolerance can

be corrected. In this paper, the average soil erosion of the study area is 3.08 $t \cdot ha \cdot yr^{-1}$, which equivalent to 41.12% of the

model, and only 15.63% in karst area.

The existing classification standards of soil erosion do not agree with the conditions of the karst region. Therefore,

it is necessary to formulate responsive classification standards according to the objective conditions in the karst region,

which not only reflects the actual situation of soil erosion accurately, but also provides a reference for the sustainable

development of the karst region. Using the ratio of theoretical soil erosion to soil loss tolerance to reflect the risk level

of soil erosion in karst areas may be a better choice.

**Acknowledgements.** This research work was supported jointly by national key research program of China (No.

2016YFC0502102 & 2016YFC0502300),"Western light" talent training plan (Class A), Chinese academy of science

and technology services network program (No. KFJ-STS-ZDTP-036) and international cooperation agency inter-

national partnership program(No.132852KYSB201 70029, No. 2014-3), Guizhou high-level innovative talent training

program "ten" level talents program(No.2016-5648), United fund of karst science research center (No. U1612441),

International cooperation research projects of the national natural science fund committee (No. 41571130074 &

41571130042), Science and Technology Plan of Guizhou Province of China (No. 2017-2966).

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
