# Peer review of "Estimation of soil erosion considering soil loss tolerance in karst area"

_Natural Hazards and Earth System Sciences, 2018_

## Referee Comment (RC1) · Anonymous Referee #1 · 29 Nov 2018

OVERALL COMMENTS: 1. I found the authors presented a nice method to describe the soil formation balance in karst areas in China which can be useful to establish the thresholds of soil tolerance, however, I feel that you need experimental data to evaluate its usefulness and to justify their conclusions. 2. Another point to consider is that soil tolerance is usually defined as a permissible limit of soil erosion which can be based on different criteria depending on the land use, environmental and ecological features and economic activities and interests. It is quite surprising that different criteria about soil tolerance used along the History are not mentioned. There is abundant literature and reviews, however, there is neither discussion nor comparison at all about the different limit which can be taken into account. In addition, the introduction is very poor, particularly if the topic of the manuscript is about the concept of tolerance (please consider

[Figure]

Smith 1941, and some reviews, among others proposed by Mccormack, et al. 1982. Current Criteria for Determining Soil Loss Tolerance; doi:10.2134/asaspecpub45.c9; Lan Li et al. 2009. An overview of soil loss tolerance. Catena 78 (2-15; 93-99); Nearing, M. et al. 2017. Natural and anthropogenic rates of soil erosion. International Soil and Water Conservation Research, 5-2, 77-84). 3. You used RUSLE to compute soil erosion for an area of 286000km2 where I was wondering if you have erosive processes different to splash, sheet and interill/rill erosion. Gullies, slides, etc must be also considered or at least discussed when you compare soil losses and formation.

DETAILED COMMENTS: 4. Abstract (page 1 line 15): please, explain how the current tolerance limits were calculated. 5. Introduction: please, see comment 2. I missed a review about the concepts of soil tolerance. In addition, I found that lines 129-140, (page 9) where you explain the problem to address should be included into the Introduction. 6. M&M: Please, mention in the text Eq. 1. The LS- factor in RUSLE is a unique factor so I suggest grouping. 7. M&M: Please, check and standardize the units in Eq. 1: for A (t.ha-1.y-1); R (MJ.mm.ha-1.h-1); K (t.h.MJ-1.mm-1.y-1) 8. Page 6, line 82, please, consider to rewrite the sentence Zhang and Fu (2003) compared the five methods. . .. Remove the reference in the final of the sentence. 9. R (MJ. Mm.ha-1.h-1), page 6, line 86. 10. Line 89 (page 2) 30 mm sounds a very high limit to consider an erosive event. In RUSLE, the thresholds to consider erosive storms were described because the computation was slower and because the small events had a small contribution to the annual value; but 30 mm in a day is a substantial storm which can be very erosive. . . 11. Page 7, line 97, please put the units of IS; be consistent with the units. 12. Table 2. You should present a complete table about the features of the study site into the chapter 2.1. Table 2 is quite difficult to interpret without a previous context of the study site. 13. Results: page 13, Line 202, please, correct "we estimated" 14. Results: page 14, line 2014, please, correct "we obtained" 15. Line 217, page 15 and abstract. See comment 4, how was calculated the tolerance to compare. 16. Discussion (line 245): how you know that you are improving the accuracy. See also comment 1.

---

## Referee Comment (RC2) · Anonymous Referee #2 · 7 Dec 2018

Overall comments The authors present an interesting method to evaluate the soil formation balance in karst areas and derive a soil erosion tolerance. It must be clearly stated that only splash, sheet and interill/rill water erosion is considered. To estimate soil erosion, soil loss tolerance is not needed; to evaluate soil erosion tolerance yes. Consider to change the work title to something like: "Estimation of soil erosion tolerance in karst area and derived evaluation/categorization of RUSLE soil losses". There are parts of the manuscript, in the methodology, results and discussion sections, that have to be improved.

Specific comments Revise word spacing; e.g. page 2 line 36 "basin(Vigiak" Homogenize soil loss units and abbreviations; e.g. page1 line l6, t•ha•yr-1, page 5 line 77 t•km-2•a-1. Be consistent also for R and K units. Keep in mind that some

values are similar, but with different units. Page 1 line 20, replace "idea" by "procedure". Page 4 line 61, specify "more than 800 mm". Page 4 line 62, "exposed karst" or "karst area"? "Exposed karst" means rock without no soil? Page 7 line 109-110, the formula was presented previously by Wishmeier (1978) –considering slope length in US customary units-. Consider refer the historical authors. Page 8 line 115-117, how is calculated the slope length values. In my knowledge and experience is a critical point, e.g. see Honghu et al 2011. Effects of DEM horizontal resolution and methods on calculating the slope length factor in gently rolling landscapes. Page 8 line 120, close } Page 9 Table 2, coefficients for paddy fields and especially for cropland must be justified. Coefficients for town, water. . . consider a mask over the area, excluding these features. Page 9 line 129, do you refer to soil loss tolerance? Page 9 line 134, "parent rock exposure" or "parent rock edafization"? Review the text from this point of view. Page 13 line 205-206, specify the limits considered to qualify erosion. Page 14, fig 2, make explicit the factor for each map. Consider using a mask to remove items where analysis is not appropriate (water, towns) and yield strange values. You must consider points 7.115 and 9.2 before working in this point. Page 14 line 210-211, "... have low soil formation rates and less soil. Therefore ... soil erosion not exceed the allowable loss of soil" do not seems coherent. Page 14 line 213, rock formation or rock exposure? Page 14 line 215, where are considered the soil loss tolerance of clastic rocs? Values seem very different. What clastic rocs can be found in the area? Page 15, fig 3, specify units and limits (see 13.205). Page 16-17, table 3. Soil erosion estimates do not change because the soil loss tolerance (T) changes. Are you considering an "effective soil" loss that could be defined as soil loss estimate minus soil loss tolerance? Review it. Specify the meaning of each table file (e.g. erosion area has not area units but a total soil loss units). Perhaps the approach would be qualify/categorize erosion with the T found, and others T values form other methods/references. 4 Discussions. Consider the notes on Table 3. Section 4.1 do not account for validity. Section 4.2. is more interesting. Page 18 line 260, some reference about experimental soil erosion data (0.4) is needed. Page 18, Table 4. Title must be "Compilation of the measured. . ."

and compare with your estimates in text. Variability must be considered. Because the main contribution of the work is the soil loss tolerance, it would be compared with other tolerance limits proposed for karst areas.

---

## Author Comment (AC1) · 24 Dec 2018

Dear reviewer: I am very grateful to your comments for the manuscript. According with your advice, we amended the relevant part in manuscript. Some of your questions were answered below. OVERALL COMMENTS: 1. In chapter 4.2, we compared the results of soil allowable loss with other studies, and proved the reasonable of the calculation of the allowable loss of soil. In addition, we also compared the soil erosion modulus corrected by the soil loss tolerance with other field or experimental data in Table 5, which proved the necessity and correctness of the correction.

2. Thank you for your comments. We have rewritten the introduction, adding a review about the concepts of soil tolerance. We have sorted out the key problems that need

to be addressed in this paper, and detailed the necessity and significance of studying the soil loss tolerance in the karst area. We increased research progress on the loss tolerance of soil, and compared the results of this paper with the current allowable loss of soil in karst areas in the chapter 4.2 of the article. The added references follows: [1] Nearing, M. A., Xie, Y., Liu, B., and Ye, Y.: Natural and anthropogenic rates of soil erosion, International Soil & Water Conservation Research, 5, 2017. [2] Mccormack, D. E., Young, K. K., and Kimberlin, L. W.: Current Criteria for Determining Soil Loss Tolerance, ASA Special Publication - American Society of Agronomy (USA), asaspecialpubli, 1982. [3] Lan, L., Du, S., Wu, L., and Liu, G.: An overview of soil loss tolerance, Catena, 78, 93-99, 2009. [4] Chai, Z.: Soil Erosion in Karst Area of Guangxi Autonomous Region, Journal of Mountain Research, 1989. 255-260, 1989. [5]Chen, X.: Research on Characteristics of Soil Erosion in Karst Mountainous Region Environment, Journal of Soil Water Conservation, 1997 [6] Wei, Q.: Soil Erosion in Karst Region of South China and its Control, Research of Soil & Water Conservation, 1996. [7] Li Y , Bai X Y , Wang S J , et al. Evaluating of the spatial heterogeneity of soil loss tolerance and its effects on erosion risk in the carbonate areas of southern China[J]. Solid Earth, 2017, 8(3):661-669. [8] Qian, Q., Wang, S. J., Bai, X., Zhou, D., Tian, Y., Li, Q., Wu, L., Xiao, J., Zeng, C., and Fei, C.: Assessment of soil erosion in karst critical zone based on soil loss tolerance and source-sink theory of positive and negative terrains, Acta Geographica Sinica, 73, 2135-2149, 2018.

3. In chapter 2.3.1,we specified that the RUSLE was used to evaluate the soil erosion in the study area, and only splash, sheet and interill/rill water erosion is considered (Liu et al., 2018)

DETAILED COMMENTS: 4. Abstract: We improve it as "Here we utilized the thermodynamic dissolution model of carbonate rocks to calculate the dissolution velocity of carbonate rocks, which was combined with the content of acid-insoluble components and the lithological characteristics to estimate the soil loss tolerance, which was then used to revise and evaluate the soil erosion in karst area."

5. Introduction: Thanks for your suggestion; we have added this section about a review about the concepts of soil tolerance to the introduction section

6. M&M: Thanks for your suggestion; LS is the topographic factor, we have grouped.

7. M&M: Thank you for your careful work. We have checked and revised the units in Eq.1.

8. Thanks for your suggestion. We have rewrite the sentence.

9. Thank you for your careful work. We have amended the units of R factor.

10. In chapter 4.1 , we specified the reason of double thresholds of rainfall erosivity in karst. Runoff does not generate in every rainfall in karst area, but mainly in heavy rain (25mm-50mm), specially rainstorm (exceeding 50mm). Because fissures are well developed in karst areas (Fig.2), most of the rainfall enters underground rivers through fissures, making the surface runoff coefficient of karst areas very low, which runoff coefficient was about 2.31%-14.72% (Wei et al., 2011). Non-karst areas generally exceed 20%, and even more than 30% (Shi et al., 2005). In view of the above, erosive rainfall threshold for the karst area is 30mm.

11. Thank you for your careful work. We have amended the units of K factor.

12. This paper refers to the previous research results and combines the local land use and agricultural activities to determine the P value (Xu et al., 2011), assigns the P factor value to the corresponding land use (Table 2), and obtains the P factor map of the study area. The obtained value is within 0–1. If the value is 0, then the area is not affected by soil erosion; if the value is 1, the area has not been subjected to any soil or water conservation measures. For the study area, paddy fields have basically been terraced, but a considerable part of the dry land has not taken any measures.

13. Thank you for your careful work. We have corrected the tense.

14. Thank you for your careful work. We have corrected the tense.

15. Equation 11 gives the formula for soil loss tolerance for different rock types in karst areas, in which the formula for calculating the maximum dissolution velocity of carbonate rocks is given in Equation 13; then the factors in Equation 13 are given in Equation 14-22.

16. In chapter4.2, we compared our result with others studies of soil loss tolerance in karst area, and proved the correctness of the calculation of the allowable loss of soil. Furthermore, we also compared the soil erosion modulus corrected by the soil loss tolerance with other field or experimental data in Table 5, which proved the necessity and reasonable of the correction.

Thank you for the kind advice. Sincerely yours, Yue Cao Corresponding author: Name: Xiaoyong Bai E-mail: baixiaoyong@126.com

Please also note the supplement to this comment:
https://www.nat-hazards-earth-syst-sci-discuss.net/nhess-2018-310/nhess-2018-310-AC1-supplement.pdf

**Supplement:**

[Figure]

Fig. 2. The geomorphologic model and actual situation of karst area

---

## Author Comment (AC2) · 24 Dec 2018

Dear reviewer: I am very grateful to your comments for the manuscript. According with your advice, we amended the relevant part in manuscript. Some of your questions were answered below. Overall comments: Dear Reviewers: Thank you for your concerning our manuscript. Those comments are all valuable and very helpful for revising and improving our paper, as well as the important guiding significance to our researches. Some of your questions were answered below. 1. Due to the congenital deficiency of the soil-forming materials, the karst area has a slow soil formation, a thin soil layer, even no soil in many areas. Actual soil holdings are much smaller than their theoretical erosion. Therefore, it is necessary to estimate soil erosion considering the soil loss tolerance.

[Figure]

2. Thank you very much for the careful review of the reviewers. Your suggestion summarizes the main research contents of this article to some extent. By carefully referring to the opinions of the reviewers, we have carefully revised the contents of the manuscript, sorted out and revised the framework of the article. The attention is on the unique geological background and geomorphological features of the karst area, and fully explained the importance of soil allowable loss assessment. Therefore, we still focus on the accurate estimation of soil erosion. We believe that your proposal has important guiding significance for our future work. In the future work, we will focus on your suggestion and hope to make new breakthroughs in soil and water conservation in the karst area.

Specific comments: 1. Thank you for your careful work. We have revised all word spacing in the text.

2. Thank you for your careful work. We have unified the units and abbreviations of soil erosion, including R factor and K factor.

3. Thanks for your suggestion. We have replaced "idea" by "procedure"

4. Thank you for your careful work. Here we refer to the research of Wei et al (2011).

5. "The exposed karst" refers to the karst outcrop area, meaning that the surface rock is carbonate rock.

6. Thanks for your suggestion. We have referred the historical authors Wishmeier et al. (1978) and Mocool et al. (1989) proposed the algorithm formula to estimate slope length factor.

7. Thank you for your suggestion. There is no doubt that the slope length factor is one of the most important factors for estimating soil erosion. The DEM resolution and the choice of the processing algorithm of different slope length factor algorithms all affect its results. In this study, because the region covers $2.86 \times 105$ km2, the highest data accuracy we can obtain is 30m for the study. The slope length algorithm refer to the

Wischmeier et al.(1978)) and Mocool et al. (1989), then the value of slope length index based on the results of Fu et al. (2015). Meanwhile, its soil layer is thin in our study area, soil conservation is very small, and there are even soilless conditions in many areas. Therefore, this paper focuses on estimating the allowable soil loss in karst area, and correcting the amount of soil erosion with it. We will consider your suggestion in the future work, thank you again.

8. This paper refers to the previous research results and combines the local land use and agricultural activities to determine the P value (Xu et al., 2011), assigns the P factor value to the corresponding land use (Table 2), and obtains the P factor map of the study area. The obtained value is within 0–1. If the value is 0, then the area is not affected by soil erosion; if the value is 1, the area has not been subjected to any soil or water conservation measures. For the study area, paddy fields have basically been terraced, but a considerable part of the dry land has not taken any measures.

9. It refers to the amount of soil loss equal to soil formation rate (Lan et al. 2009).

10. Thank you for your careful work. We have changed it to "When the rate of soil erosion is greater that of soil formation of the parent rock."

11. According to classification and gradation standards of soil erosion (SL190-2007, 2008), we divided soil erosion into six grades, which are micro-degree (A≤5 tÂůha-1Âůyr-1), mild (5 tÂůha-1Âůyr-1<A≤25 tÂůha-1Âůyr-1), moderate (25 tÂůha-1Âůyr-1<A≤50 tÂůha-1Âůyr-1), strong (50 tÂůha-1Âůyr-1<A≤80 tÂůha-1Âůyr-1), pole strong (80 tÂůha-1Âůyr-1<A≤150 tÂůha-1Âůyr-1) and violent (A>150 tÂůha-1Âůyr-1).

12. Thank you for your suggestion. We have identified the factor for each map in Fig.3 (Fig.2 before). In addition, we use mask to remove water, towns and other areas that do not produce soil erosion, and the soil erosion modulus is no data.

13. Thank you for your careful work. We have rewritten this sentence as follows: "Compared with non-karst areas, the karst area with wide carbonate rocks has the

characteristics of low soil formation rate and less soil. Therefore, the actual erosion of soil in the karst area will not exceed the allowable loss of soil."

14. Thank you for your careful work. We have changed this to "the soil formation rate of other rock types".

15. Thank you for your careful work. This refers to non-carbonate rock, we have changed the full text to other rock types, the value of soil formation rate is determined according to classification and gradation standards of soil erosion (SL190-2007, 2008).

16. Thank you for your suggestion. We have already indicated the unit of soil loss tolerance Figure 4-a, and Fig.4-b is graded according to classification and gradation standards of soil erosion (SL190-2007, 2008)

17. First, we estimate soil erosion based on RUSLE model, but the soil formation rate is slow and the soil layer is thin. The actual soil formation is much smaller than the theoretical erosion amount, which is not in line with objective reality. Therefore, we estimate the rock weathering rate in the carbonate rock area as the maximum threshold of soil loss tolerance, and replace the soil erosion with the soil loss threshold in the area where the soil erosion in theory exceeds that.

18. Thank you for your careful work. I am sorry that this is a mistake in our work. We have changed the unit of erosion area to ha.

19. Discussions. In chapter 4.1, we specified the reason of double thresholds of rainfall erosivity in karst. In chapter4.2, we compared our result with others studies of soil loss tolerance in karst area, and proved the correctness of the calculation of the allowable loss of soil. Furthermore, we also compared the soil erosion modulus corrected by the soil loss tolerance with other field or experimental data in Table 5 (chapter4.2), which proved the necessity and reasonable of the correction.

20. Thank you for your suggestion. We have changed the title of table 5 to "Comparison with the Experimental results", in which, we compared other field or experimental data

with our estimates in text of soil erosion modulus, proving the reasonable of the our correction.

We would like to express our great appreciation to you for comments on our paper. Sincerely yours, Yue Cao Corresponding author: Name: Xiaoyong Bai E-mail: baixiaoyong@126.com

Please also note the supplement to this comment:
https://www.nat-hazards-earth-syst-sci-discuss.net/nhess-2018-310/nhess-2018-310-AC2-supplement.pdf

**Supplement:**

[Figure]

Fig. 2. The geomorphologic model and actual situation of karst area